# The conditional Fama-French model and endogenous illiquidity: A robust instrumental variables test

**François-Éric Racicot** [1,2☯] *, **William F. Rentz** [1☯], **David Tessier** [3☯], **Raymond Théoret** [4,5☯]

**1** Telfer School of Management, University of Ottawa, Ottawa, ON, Canada, **2** Affiliate Research Fellow, IPAG Business School, Paris, France, **3** Département des Sciences Administratives, Université du Québec en Outaouais (UQO), Gatineau, QC, Canada, **4** Ecole des Sciences de la Gestion, Université du Québec à Montréal (ESG-UQAM), Montréal, QC, Canada, **5** Chaire d'information Financière et Organisationnelle, ESG-UQAM, Montreal, QC, Canada

☯ These authors contributed equally to this work.
\* Racicot@telfer.uottawa.ca

**Data Availability Statement:** The data are publicly available. Our sample ranges from January 1968 through December 2016. The 12-sector portfolio returns and the market risk factors are drawn from the French's website (https://mba.tuck.dartmouth.

## Abstract

We investigate *conditional* specifications of the five-factor Fama-French (FF) model, augmented with traditional illiquidity measures. The motivation for this time-varying methodology is that the traditional static approach of the FF model may be misspecified, especially for the endogenous illiquidity measures. We focus on the time-varying nature of the Jensen performance measure $\alpha$ and the market systematic risk sensitivity $\beta$, as these parameters are essentially universal in asset pricing models. To tackle endogeneity and other specification errors, we rely on our robust instrumental variables (RIV) algorithm implemented via a GMM approach. In this dynamic or time-varying *conditional* context, we generally find that the most significant factor is the market one, but illiquidity may matter depending on which states or estimation methods we consider. In particular, sectors whose returns embed a market illiquidity premium are more exposed to a binding funding constraint in times of crisis, which leads to deleveraging and a resulting decrease in systematic risk.

## Introduction

We cast the five-factor Fama-French (FF) model [1,2], which features static parameters, into the conditional framework of Ferson and Schadt [3] and others [4,5,6,7,8] to account for the time-varying nature of the standard measures of performance $\alpha$ and market risk premium exposure $\beta$. We focus on these two parameters because there are widely used by finance academics and practitioners. [9] further discusses this model in the context of the business cycle and the subprime crisis but without venturing into the time-varying dynamics of parameters. Our motivation for a *conditional* framework is the well-known fact that the static alpha and beta parameters of the CAPM [10,11,12] or the extended FF model may be misspecified [3,4,5,6,7,8] because they neglect the time-varying features of alpha and beta. [13] expresses some doubt about whether a Ferson and Schadt *conditional* model for time-varying

edu/pages/faculty/ken.french/data_library.html).
The term spread and the VIX are provided by the
database managed by the Federal Reserve Bank of
St. Louis (https://fred.stlouisfed.org/). The IML
illiquidity measure comes from the Pastor's
database (https://faculty.chicagobooth.edu/lubos.
pastor/research/). The authors did not have any
special access privileges to the data.

**Funding:** Financial support from the IPAG Business
School, Paris, France (https://www.ipag.edu/en)
and the Social Sciences and Humanities Research
Council (SSHRC) of Canada, grant no. 435-2019-
0078 (http://www.sshrc-crsh.gc.ca/home-accueil-
eng.aspx) is acknowledged. Recipient name:
François-Éric Racicot. No other funding was
obtained for this research.

parameters actually performs better than a static parameter model. Nevertheless, [14,chap.11] compares recursive/rolling regressions with the optimal Kalman filtering [15] approach and concludes that the Kalman filter is more accurate in terms of the usual measure of forecast errors but not significantly better. Consequently, we accept the idea that the recursive/rolling regression approach is similar in spirit to the Ferson and Schadt *conditional* model.

We focus on the time-varying nature of the Jensen [16] performance measure $\alpha$ and the market systematic risk exposure $\beta$, since these parameters are essentially universal in asset pricing models and cost of capital estimates (e.g., [17]). The coefficients (sensitivities) of the other risk factors (premiums)—i.e., size effect *SMB*, value effect *HML*, profitability effect *RMW*, investment policy effect *CMA*, and augmented with the Pástor and Stambaugh (PS) [18,19] tradable illiquidity effect *IML*—will not be made time-varying in our modeling approach since they are secondary parameters. However, we test for their significance in each portfolio considered in order to determine if factors other than the market one have significance to explain asset returns in our time-series approach.

The five-factor FF model [1] adds the profitability effect *RMW* and the investment policy effect *CMA* to their original three-factor model. These two factors are related to Tobin's [20] q, which was further developed in [21–24] and known as in the Cochrane's q theory. Tobin's q is the ratio of the market value of the firm over the replacement cost of assets [17], which is often proxied by the firm's market-to-book value. Note the analogy of the last definition with the price-to-book value per share (P/B) ratio, which stands as a convenient proxy for Tobin's q. However, use of a proxy may entail measurement errors. In this regard, Tobin' q is notoriously known to be measured with errors (e.g., [25]). Furthermore, like many other variables and financial ratios [26], Tobin's q is highly skewed. Thus, as another empirical issue, the normal distribution may not be applicable.

In addition to these potential measurement errors in Tobin's q, the illiquidity measure that we rely on—PS [18,19] *IML*—is also a proxy. Therefore, measurement errors are also expected for illiquidity. In fact, the original gamma version of the PS factor may be seen as a generated variable as it is obtained from a regression. In the econometric literature [27–29], researchers warn that these kinds of specification/measurement errors may result in biased inference and, more seriously in specific cases, in inconsistency. Furthermore, according to [30], traditional liquidity measures are endogenous (e.g., in our case the PS measure), which therefore produces an endogeneity bias when estimating parameters with OLS. In addition, the well-known CAPM risk premium cannot be perfectly measured and is usually proxied by a large-cap portfolio such as the S&P 500 index [31,32]. Efforts to address these measurement problems are therefore on point.

In this article, we first aim to contribute to the applied financial economics (i.e., financial modeling) literature by proposing a robust instrumental variables (RIV) algorithm cast into a time-varying *conditional* model. The latter can be implemented in a GMM framework to correct for endogeneity and potential misspecification and measurement errors. To the best of our knowledge, we believe that we are the first authors to make this investigation. Secondly, we augment the FF [1,2] five-factor model with an illiquidity measure and transpose it to our time-varying framework. This will allow us to test the performance measure $\alpha$ and systematic risk $\beta$ exposures in a time-varying context while accounting for cyclicality and other dimensions of risk, including illiquidity.

The RIV that we propose in a time-varying GMM setting are based on higher moments of the observed variables. These instruments appear to be strong (i.e., not weak) and respond to the concerns of several researchers (e.g., [33,34,35,36]) that the two-stage least squares (TSLS) estimator may be inconsistent when using weak instruments. Furthermore, [37] discuss the fact that weak instruments may increase the variance of the IV estimator. Our instruments are

based on an optimal combination of the [38,39] estimators following the framework of [40]. This optimal combination may account for heteroscedasticity, which leads to a consistent estimator even when measurement errors are considered [41]. However, to deal with both heteroscedasticity and/or autocorrelation, the [36,42] HAC (heteroskedasticity and autocorrelation consistent) matrix is used as a weighting matrix, and, therefore, the GMM estimator seems appropriate. We propose to extract the aforementioned RIV and use them in the time-varying GMM framework.

This article is organized as follows. The next section describes the five-factor augmented time-varying *conditional* version of the FF [1,2] model. Then we discuss our proposed robust instrumental variables (RIV) algorithm implemented via a GMM framework referred to as RIV GMM, and we report our results. Finally, we set out our conclusion and propose suggestions for future research.

## Five-factor augmented Fama-French *conditional* model

The five-factor FF [1,2] model may be written as

$$r_{it} - r_{ft} = \alpha_i + \beta_{1i}[r_{mt} - r_{ft}] + \beta_{2i}SMB_t + \beta_{3i}HML_t + \beta_{4i}RMW_t + \beta_{5i}CMA_t + \varepsilon_{it} \qquad (1)$$

where $r_{it} - r_{ft}$ is the excess return on the i[th] FF sector portfolio observed for period $t$, $r_{mt} - r_{ft}$ is the excess return on the market portfolio, $SMB_t$ is the return on the small minus big capitalization portfolio representing the size effect, $HML_t$ is the return on the high book-to-market minus the low book-to-market portfolio representing the value effect, $RMW_t$ is the robust minus weak profitability portfolio, $CMA_t$ is the conservative minus aggressive investment policy portfolio, and $\varepsilon_{it}$ is the error term which might be autocorrelated and/or conditionally heteroscedastic (i.e., ARCH process).

The advantage of the five-factor FF model over the earlier three-factor model is the theoretical justification developed for the two new factors $RMW_t$ and $CMA_t$ based on Tobin's [20] q cast in the investment function $I = f(q)$. In fact, solving a dynamic programming problem via the Bellman's [43] principle of optimality, Abel [44] obtains,

$$I = [q/\theta r]^{1/[\theta-1]} \qquad (2)$$

where $q$ is the expected marginal revenue product of capital, $\theta$ is the constant elasticity of the cost of investment, and $r$ is the interest rate. More recently, Chow ([45],chap.8,p.176) developed the Abel model using Lagrangian multipliers instead of the Bellman equation. [46] derive an analytic solution for Tobin's q based on a stochastic dynamic framework using the classical geometric Brownian motion. [47] write the conditional expected return as a positive function of conditional expected profitability and a negative one with respect to investment, as the firm's rate of return declines with increasing investment as it slides down on the investment opportunity curve or schedule (IOS).

Our augmented FF model that includes illiquidity *IML* may be written as

$$r_{it} - r_{ft} = \alpha_i + \beta_{1i}[r_{mt} - r_{ft}] + \beta_{2i}SMB_t + \beta_{3i}HML_t + \beta_{4i}RMW_t + \beta_{5i}CMA_t + \beta_{6i}IML_t + \varepsilon_{it} \ (3)$$

where *IML* is the return of a portfolio long on illiquid and short on liquid securities. We use the PS [18,19] tradable liquidity as—i.e., the level of aggregated liquidity (gamma), and the innovations in aggregate liquidity a proxy for this variable. Note that PS [18] have developed two other liquidity measures—i.e., the level of aggregated liquidity (gamma), and the innovations in aggregate liquidity. However, in an asset pricing model, all the explanatory variables must be tradable [18,19]. Among the three liquidity variables designed by PS [18,19], only *IML*

is tradable while the two others result from regressions. Introducing non-tradable variables in an asset pricing model should bias the alpha.

The variable IML might be considered as a generated variable, therefore resulting in biased inference from the OLS estimator [27–29]. As discussed in the section below on endogeneity, traditional liquidity measures, such as one based on PS [18,19], may be seen as endogenous variables [30]. The endogeneity bias must therefore be handled in some fashion. In this regard, we propose a parsimonious approach that has the virtue of being robust to several types of specification errors. Note that a good econometric model should only include relevant variables and not redundant ones. Parsimonious specifications lead to better forecasts and are less prone to overfitting problems [14]. In this regard, information criteria—like Akaike and Schwarz criteria—include a function that penalizes the number of variables included in an econometric model.

## Conditional augmented Fama-French five-factor model

The literature has established that the static nature of the CAPM or the extended FF model may be impregnated with specification errors [3,4,5,6]. Recent applications of conditional models, for instance, include pension fund performance and the analysis of the impact of the informational content of extra-financial performance scores on systematic risk (e.g., [48,49]). However, none of these studies provide specification error tests or discuss possible endogeneity issues.

Note that Ghysels [13] expresses some concern about whether a Ferson and Schadt *conditional* model for time- varying parameters actually performs better than a static parameter model. Nevertheless, we believe that conditional models, because of their dynamic nature, may help in shedding light on some well-known financial puzzles (e.g., the $\alpha$ puzzle) even though these models might not qualify as being optimal in the Kalman [15] filter sense. Support for this approach comes from Ghysels and Marcellino [14,chap.11] who discuss several measures of comparison based on out-of-sample forecasting and compare the rolling/recursive regression approach to the time-varying parameters model that they estimate with the Kalman filter. Therefore, we adopt the *conditional* model general formulation of our augmented FF six-factor model because of its parsimonious features. In this formulation, the unconditional $\alpha$ and $\beta$ in (3) become the *conditional* $\alpha$ and $\beta$ in (4) and (5), respectively,

$$\alpha_{it}(\Omega_{0,t-1}) = \alpha_{i0} + \varphi\Omega_{0,t-1} \tag{4}$$

$$\beta_{1it}(\Omega_{1,t-1}) = \beta_{1i0} + \omega\Omega_{1,t-1} \tag{5}$$

where $\alpha_{it}$ is a function of the information set $\Omega_{0,t-1}$ which is a matrix of explanatory variables for our portfolios i = 1 to N and for time periods ranging from t = 1 to T, and $\varphi$ is its corresponding vector of coefficients. Similarly, $\beta_{it}$ is a function of the information set $\Omega_{1,t-1}$ which is another matrix of explanatory variables for the same portfolios and time periods, and $\omega$ is its corresponding vector of coefficients.

Substituting the *conditional* alpha and beta (4) and (5), respectively, into (3) yields

$$\begin{aligned} r_{it} - r_{ft} &= \alpha_{i0} + \varphi\Omega_{0,t-1} + \beta_{1i0}[r_{mt} - r_{ft}] + \omega\Omega_{1,t-1}[r_{mt} - r_{ft}] \\ &+ \beta_{2i}SMB_t + \beta_{3i}HML_t + \beta_{4i}RMW_t + \beta_{5i}CMA_t + \beta_{6i}IML_t + \varepsilon_{it} \end{aligned} \tag{6}$$

In writing the *conditional* formulation of our model (6), the approach taken here expresses some parameters in terms of past information. In our case, we specify $\alpha$ and $\beta$ *conditional* on past information. This is analogous to [49]. By specifying the information sets $\Omega_{0,t-1}$ and $\Omega_{1,t-1}$,

we obtain the implementable *conditional* version of the augmented FF model by transforming (6) into

$$
\begin{aligned}
r_{it} - r_{ft} = \alpha_{it}(\Omega_{0,t-1}) + \beta_{1it}(\Omega_{1,t-1})[r_{mt} - r_{ft}] + \beta_{2i}SMB_t \\
+ \beta_{3i}HML_t + \beta_{4i}RMW_t + \beta_{5i}CMA_t + \beta_{6i}IML_t + \varepsilon_{it}
\end{aligned}
\tag{7}
$$

with

$$
\alpha_{it}(\Omega_{0,t-1}) = \alpha_{0i} + c_{1i}[r_{mt-1} - r_{ft-1}] + c_{2i}[spread_{t-1}]
\tag{8}
$$

$$
\beta_{1it}(\Omega_{1,t-1}) = \beta_{0i} + c_{3i}[r_{mt-1} - r_{ft-1}] + c_{4i}[spread_{t-1}] + c_{5i}[IML_{t-1}]
\tag{9}
$$

where $\Omega_{0,t-1}$ is a matrix information set at period t-1 consisting of 2 components which are $[r_{mt-1} - r_{ft-1}]$, the excess market return and the term spread $[spread_{t-1}]$, the spread between the ten-year Treasury constant maturity rate and the 90-day Tbill rate, and where $\Omega_{1,t-1}$ is a matrix information set at period t-1 consisting of 3 components with the first two identical to the components in $\Omega_{0,t-1}$ and the third component being $IML_{t-1}$—i.e., the illiquidity risk premium. When proceeding with estimation of the coefficients in (8) and (9), the portfolios once more run from i = 1 to N and time periods from t = 1 to T. *VIX*—the S&P 500 implied volatility index—is a cyclical variable that, like *IML*, may capture the change in uncertainty in the market. It is also known as the investor fear gauge. However, in this article, we mainly focus on illiquidity. Therefore, we report results only for illiquidity in our time-varying estimates for beta. The preliminary results for *VIX* are interesting but outside the scope of this article.

## Endogeneity issue with traditional liquidity measures

Broadly, liquidity is defined as the cost of exchanging assets for cash. Since this cost is related to the behavior of market makers, Adrian et al. [30] argue that traditional liquidity measures are endogenous. This implies that we can write our augmented FF model, for instance, as a simultaneous two-equation model

$$
\begin{aligned}
r_{it} - r_{ft} = \alpha_i + \beta_{1i}[r_{mt} - r_{ft}] + \beta_{2i}SMB_t + \beta_{3i}HML_t + \beta_{4i}RMW_t + \beta_{5i}CMA_t + \beta_{6i}IML_{it} \\
+ \varepsilon_{it}
\end{aligned}
\tag{10}
$$

$$
IML_{it} = f(\sigma_{it}; Dealers\ models_{it}; Shock/no\ shock_{it})
\tag{11}
$$

where $IML_{it}$ is a nonlinear function of return volatility $\sigma_{it}$ and possibly a linear function of the other two variables. Moreover, its response to volatility is assumed asymmetric: moderate increases in $\sigma_{it}$ may in fact improve liquidity while large increases have the reverse effects. The dealers models variable in (11) refers to the fact that dealers have been changing their business model over time, from a principal to an agency one [30]. More precisely, dealers have moved from a principal model, where they hold an inventory of assets, to an agency model in which they hold no inventory. This makes it possible to shift the inventory risk to the investors while at the same time creating a narrower bid-ask spread, which necessarily improves liquidity. The impact of this variable on liquidity could thus be positive or negative. The same logic applies to the shock/no shock variable. A shock event might induce liquidity to be endogenously created as investors with different expectations, objectives, options (financial or real) and hedging strategies, constraints, and risk tolerances enter the market to meet their trading needs. Thus, liquidity may improve as buyers and sellers enter in the market simultaneously. In a no-shock period, investors may wait to transact, reducing volume, which would therefore affect

negatively traditional liquidity measures even if intrinsic liquidity itself has not been adversely affected.

As previously mentioned, one of our main objectives is to propose a parsimonious way to tackle the problem of endogeneity without relying on a second equation such as (11). This is the subject of the next section. Note also that [30] contend that liquidity (*LIQ*) could be modeled via a continuous Gaussian model with infrequent jumps. We could therefore translate this suggestion further by assuming a basic model such as [50] or [51] as follows

$$dLIQ = (r_f - \lambda k)dt + \sigma dz + dP \qquad (12)$$

which, after a typical Euler discretization, $dP$ or $\Delta P$, could be a Poisson jump process with $\lambda$, the average number of jumps per year and $k$, the average jump size measure as percentage of asset price.

## Robust IV estimation algorithm of the augmented Fama-French (FF) conditional model

The RIV GMM algorithm discussed in this section is implemented with the EViews programming matrix language. As is well documented, the main problem with the GMM estimator lies in the choice of instruments. As discussed in [37] and further analysed in [41], the instruments chosen should be robust (not weak). Weak instruments can increase the variance of the resulting estimator or worse, may yield inconsistent estimates. Finding instruments is not such an easy task. Lagged values of the explanatory variables are widely used as instruments (e.g., [37]). Nevertheless, economic theory should be the framework used to guide the selection of instrumental variables.

Essentially, the robust instrumental variables (RIV) method we develop here in the context of the *conditional* FF model, relies on the generalized method of moments (GMM), which we refer to as RIV GMM. We do not resort to lagged values as usually recommended. Instead, we employ average contemporaneous values taken to a power of 2 and 3 expressed in a deviation of the mean in order to reduce the erratic behavior of higher moment instruments. More precisely, we use a weighted average of the powers obtained via generalized least squares (GLS). This average can be considered optimal in the Aitken sense. In this fashion, once the instruments are obtained, they can be incorporated directly into the GMM estimator relying on a weighting matrix. The usual matrix choices are the optimal HAC [36], [42] with automatic lag selection, Hansen [52] or White [53] matrices. More precisely, the HAC procedure provides consistent matrix estimators $\hat{\Omega}$ given by $\hat{\Omega} = \sum_{j=1}^{T-1} \omega_{j,T}\hat{\Gamma}_{j,T} = \sum_{j=1}^{T-1} k(j/b_T)\hat{\Gamma}_{j,T}$, with $\hat{\Gamma}_{j,T}$ equal to the sample autocovariances. In this equation, $k$ is the kernel function and $b_T$ is the bandwith. The kernel function is used to determine the weights $\omega_{j,T}$ in $\hat{\Omega}$ [14]. There exists a large number of kernel functions—e.g., Bartlett, Daniel and Parzen. The bandwith is a positive value parameter which determines how many $\hat{\Gamma}_{j,T}$ are included in the HAC estimator. It determines the maximum number of lags used to compute this estimator. For instance $k(j/b_T) = 0$ for $|j| > b_T$. In this regard, a popular choice for the bandwith selection is the method proposed by Newey-West [36,42].

Our contribution is therefore to further develop the methodology discussed in [41] and in the context of our time-varying model of risk exposures. The methodology developed in [41,54] is static. The algorithm discussed in these previous works is therefore extended to the time-varying application developed in this article. However, rather than presenting the algorithm in a hermetic way as is done in the aforementioned articles, we describe it below in a

more parsimonious way, which may enhance the comprehension of the methodology and give rise to new applications.

The gist of our algorithm is as follows. To compute the parameters in (8) and (9), first substitute these two equations into (7), which creates cross-products for some of these variables. Then apply OLS or RIV GMM on (7), thereby creating a model with 12 parameters to estimate. Finally, to compute the profile of the time-varying alpha and beta given by (8) and (9), substitute the estimated coefficients of the OLS or RIV GMM regression, which must be multiplied by the observed values of the independent variables. The result is the desired time series of *conditional* parameters that we later plot to obtain $\alpha$ and $\beta$ profiles.

We can now probe deeper into our algorithm. The methodology suggested when one or all the variables are endogenous (e.g., illiquidity in our case) may be illustrated with a simple linear regression with only one explanatory variable

$$y_t = c + \beta x_t + \varepsilon_t \tag{13}$$

where $x_t$ is an explanatory variable that may be endogenous, and $\varepsilon_t$ is the innovation. As previously explained, the methodology we propose to tackle endogeneity is based on a weighted average of the squared and third power deviations of $x_t$ as instruments. This instrumental variables (IV) approach may be implemented on $x_t$ via OLS

$$x_t = a_0 + a_1(x_t - \bar{x})^2 + a_2(x_t - \bar{x})^3 + e_t \tag{14}$$

to obtain the predicted value $\hat{x}_t$. To tackle the endogeneity problem, the next step is to substitute this predicted value into (13)

$$y_t = c + \beta \hat{x}_t + \varepsilon_t \tag{15}$$

Or, by replacing the predicted value of $x_t$ by its value in term of instruments

$$y_t = c + \beta[\hat{a}_0 + \hat{a}_1(x_t - \bar{x})^2 + \hat{a}_2(x_t - \bar{x})^3] + \varepsilon_t \tag{16}$$

and run OLS on (16) or equivalently on (15). Note that (16) clearly highlights the fact that this regression is a nonlinear one because of the product of parameters and could be efficiently estimated in only one step via nonlinear least-squares (NLS).

However, our suggested approach is more akin to two-stage least squares (TSLS). This idea may be expressed parsimoniously using standard econometrics (e.g., [37]), that may provide more intuition of the methodology. Assume that $z$, the instrument, is based on the predicted value: $[\hat{a}_0 + \hat{a}_1(x_t - \bar{x})^2 + \hat{a}_2(x_t - \bar{x})^3]$. More precisely, the TSLS estimator here can be described via two separate regressions. The first one is to consider a regression of the form

$$y = x\beta + \varepsilon \tag{17}$$

where $x$ is regressed on $z$

$$x = z\alpha + e \tag{18}$$

to obtain the predicted value of x which yields the following IV estimator after, of course, replacing the predicted value of (18) in (17) and running OLS on the result

$$\hat{\beta} = (\hat{x}'\hat{x})^{-1}\hat{x}'y = (x'(z(z'z)^{-1}z'x))^{-1}(x'z(z'z)^{-1}z'x)y = \hat{\beta}_{TSLS} \tag{19}$$

(19) is in the format of the well-known TSLS estimator augmented with our robust instrumental variables (RIV). Now that we have (19), we can express that result in the more general

GMM format, which can be stated simply as

$$q = \bar{G}'W\bar{G} \qquad (20)$$

where $W$ is the weighting matrix to which we previously alluded, $\bar{G} = (1/n)G_n = (1/n)\sum_{i=1}^{n} g_i$ the moment conditions (i.e., the $m$ moments conditions $E[g_i(\beta)] = 0$ that the data generating process is assumed to satisfy) which in our case is the suggested RIV, and $q$ is a quadratic function to be minimized with respect to the required parameters estimation. The Appendix further discusses our algorithm showing how to compute a weighted average of the squared and third power deviations of the explanatory variables via generalized least squares (GLS).

Note that the primary version of the suggested RIV does not include lagged values of the explanatory variable, as is often suggested to confront endogeneity. However, nothing prevents the researcher from including those along with the suggested instruments. This in turn will yield an overidentified system (20), and the Sargan [55–57] test or its equivalent J-test can be performed to analyse the identification issue [37]. In its original version, our approach has the virtue of not requiring this kind of testing since it is exactly identified.

## Empirical results

### Descriptive statistics

Our sample ranges from January 1968 through December 2016. The 12-sector portfolio returns and the market risk factors—i.e., $R_m$-$R_f$, *SMB*, *HML*, *CMA*, and *RMW*—are drawn from the French's website (https://mba.tuck.dartmouth.edu/pages/faculty/ken.french/data_library.html). The term spread and the VIX are provided by the database managed by the Federal Reserve Bank of St. Louis (https://fred.stlouisfed.org/). The *IML* illiquidity measure comes from the Pastor's database (https://faculty.chicagobooth.edu/lubos.pastor/research/).

Table 1 provides the descriptive statistics for the monthly excess returns for each of the Fama-French portfolios of twelve sectors and for the 12-sector average for the period January 1968 through December 2016. The mean return is close to 1% in our sample, albeit with some minor variation across sectors. The average standard deviation is somewhat over 4% with negative skewness and kurtosis well above 3. However, a minor word of caution should be noted when analyzing statistics such as skewness and kurtosis—i.e., kurtosis is related to skewness, *viz.*, kurtosis $\geq$ skewness$^2$ + 1 (e.g., [58,59]). Not surprisingly, the average Jarque-Bera (*JB*) statistic [60], which is asymptotically chi-squared distributed with 2 degrees of freedom with normality as the null hypothesis, is well-above the 1% level of 9.21, ranging from a low of 24.93 for energy to a high of 551.83 for durables. Most sectors show first-order serial correlation. Return autocorrelation may be attributable to illiquid portfolios—like the one associated with the durable sector—or to income smoothing as in the money sector. Care must be taken when estimating regressions with such data. The estimator that we propose in this article is based on cross-sample higher moments. Therefore, the fact that there is substantial kurtosis might be seen as another argument in favor of our robust instruments. In addition, because we transpose our instruments into a GMM setting, all the aforementioned nonspherical issues should be addressed.

Table 2 displays the descriptive statistics for the factors used in our *conditional* model. The range of these *JB* statistics is somewhat larger than the range for the sector returns, from a low of 21.82 for the illiquidity factor *ILM* to a high of 3630.35 for the profitability factor *RMW*. Nevertheless, all of these *JB* statistics greatly exceed the 1% level, leading to a rejection of the null hypothesis of normality.

**Table 1. Descriptive statistics for the Fama-French sector monthly returns (1968/01 to 2016/12).**

|  | Mean | Median | Max | Min | AR(1) | SD | Skew | Kurt | JB |
|---|---|---|---|---|---|---|---|---|---|
| 1 Nodur | 1.08 | 1.08 | 18.88 | -21.03 | 0.15*** | 4.34 | -0.27 | 5.03 | 107.96*** |
| 2 Durbl | 0.85 | 0.83 | 42.63 | -32.63 | 0.13*** | 6.41 | 0.12 | 7.74 | 551.83*** |
| 3 Manuf | 0.95 | 1.16 | 21.08 | -28.58 | 0.10*** | 5.36 | -0.49 | 5.62 | 191.78*** |
| 4 Enrgy | 1.02 | 0.93 | 24.56 | -18.33 | 0.02 | 5.59 | 0.04 | 4.13 | 31.51*** |
| 5 Chems | 0.93 | 1.04 | 20.22 | -24.59 | 0.05 | 4.69 | -0.22 | 5.18 | 121.14*** |
| 6 Buseq | 0.89 | 0.83 | 20.76 | -26.03 | 0.07* | 6.60 | -0.19 | 4.24 | 40.90*** |
| 7 Telcm | 0.95 | 1.16 | 21.36 | -16.36 | 0.08* | 4.74 | -0.24 | 4.19 | 40.34*** |
| 8 Utils | 0.88 | 0.93 | 18.84 | -12.65 | 0.08* | 4.10 | -0.10 | 3.99 | 24.93*** |
| 9 Shops | 1.03 | 0.97 | 25.86 | -28.23 | 0.17*** | 5.26 | -0.26 | 5.41 | 149.14*** |
| 10 Hlth | 1.04 | 1.09 | 29.52 | -20.46 | 0.03 | 4.94 | 0.06 | 5.46 | 148.41*** |
| 11 Money | 1.02 | 1.36 | 21.10 | -22.10 | 0.15*** | 5.58 | -0.41 | 4.59 | 78.02*** |
| 12 Other | 0.78 | 1.04 | 19.36 | -29.26 | 0.14*** | 5.48 | -0.48 | 5.17 | 137.35*** |
| Average | 0.95 | 1.13 | 16.19 | -21.97 | 0.12*** | 4.35 | -0.45 | 5.08 | 126.51*** |

The significance levels of the p-values are 1% indicated by***; 5%, **; and 10%, *.

AR(1) is an autoregressive model of order 1. JB is the Jarque-Bera test for normality.

## Discussion of *IML* vs term spread

In this article, we concentrate on the Pástor-Stambaugh (PS) tradable liquidity measure because it appears to have been adopted by financial practitioners [19,17]. This measure, which we denote by *IML*, is an illiquidity risk premium as it gauges the difference in returns of a portfolio of illiquid securities and a portfolio of liquid securities. Thus, in quiescent periods, we would expect to earn an illiquidity risk premium; whereas, in times of turmoil the liquid portfolio should outperform the illiquid one.

Goyenko et al. [61] examine many different liquidity measures, including the tradable liquidity measure (IML) and its non-tradable version (gamma), both due to Pástor and Stambaugh [18]. They provide a better score to the gamma measure. However, for the estimation of the FF model the factors ought to be expressed in risk premia. To the best of our knowledge, IML is the only available replicating portfolio which accounts for illiquidity.

**Table 2. Conditional Fama-French model risk factors 1968 m01-2016 m12.**

|  | Mean | Median | Max | Min | Std. Dev. | Skewness | Kurtosis | Jarque-Bera |
|---|---|---|---|---|---|---|---|---|
| *Spread* | 1.67 | 1.82 | 4.42 | -2.65 | 1.26 | -0.52 | 2.83 | 27.69*** |
| *IML* | 0.40 | 0.35 | 11.08 | -12.89 | 3.37 | -0.02 | 3.94 | 21.82*** |
| *VIX* | 19.71 | 18.01 | 59.89 | 10.42 | 7.48 | 1.72 | 7.55 | 439.43*** |
| $R_m$-$R_f$ | 0.49 | 0.81 | 16.10 | -23.24 | 4.54 | -0.51 | 4.79 | 103.88*** |
| *SMB* | 0.19 | 0.04 | 18.27 | -14.85 | 3.06 | 0.40 | 6.37 | 294.37*** |
| *HML* | 0.38 | 0.30 | 12.90 | -11.10 | 2.90 | 0.06 | 4.96 | 94.75*** |
| *RMW* | 0.26 | 0.28 | 13.51 | -18.72 | 2.29 | -0.32 | 15.16 | 3630.35*** |
| *CMA* | 0.35 | 0.22 | 9.58 | -6.88 | 2.02 | 0.33 | 4.58 | 71.59*** |

The significance levels of the p-values are 1% indicated by***; 5%, **; and 10%, *. The Jarque-Bera (*JB*) statistic is a test for normality. The sample totals 588 observations for all factors except for *VIX* which is 324 over 1990 m01-2016 m12. *Spread* is the difference between the ten-year Treasury constant maturity rate and the 90-day Tbill rate available from the St. Louis Federal Reserve FRED database. *IML* is the Pástor-Stambaugh tradable liquidity risk factor available the from Pastor's website. The implied volatility index *VIX* is available from the CBOE database. The other risk factors come from French's website.

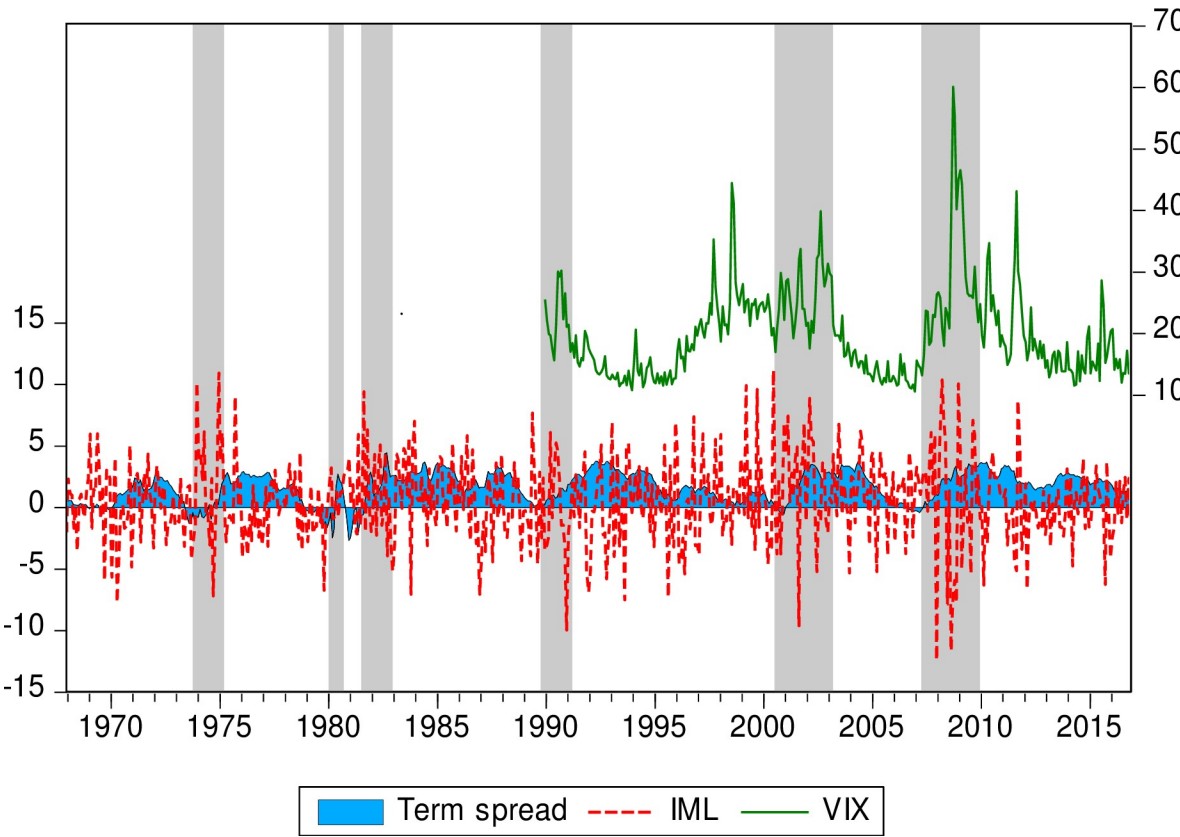

**Fig 1. Pástor-Stambaugh liquidity, term spread, and VIX.** Notes: Shaded areas represent US recessions or crises. *IML*, a portfolio of illiquid minus liquid stocks, is the Pástor-Stambaugh tradable liquidity measure and term spread is the yield spread between 10-year T-bonds and 3-month T-bills.

We also use the term spread—defined as the difference between the federal 10-year rate and the 90-day Tbill rate—as a funding liquidity measure [62]. In Fig 1, the shaded areas represent U.S. recessions. Generally, when the term spread is increasing, the volatility of the Pástor-Stambaugh measure tends to rise. In the subprime crisis (2007–2009), the term spread is high and the volatility of *IML* is also high but seems to be asymmetric in character: There are more important negative values than positive ones for *IML*, which leads to a cumulative negative return for the *IML* portfolio. In fact, the negative values of this measure are the largest in magnitude in our sample during the crisis. Note, for instance, that this observation is in line with Nelson's [63] celebrated EGARCH model of Black's [64] leverage effect on the asymmetry of the impact of bad news versus good news on stock volatility.

The term spread is widely accepted in macroeconomics as a leading indicator of the future state of the economy. When the term spread increases, it is anticipated that the economy will recover and vice a versa [65]. In this regard, when the term spread peaks, market participants begin to expect a decrease in long-term interest rates, which contributes to revive confidence and foster interest-sensitive expenditures like investment and the consumption of durables goods. This is the expectation effect associated with the term spread. Conversely, a flattening or inverted yield curve (negative term spread) is a strong indicator of an upcoming recession. Furthermore, it appears that the term spread can be seen as a predictor of PS future volatility. In other respects, implied volatility indices such as the *VIX* can be considered as a forward-looking view of volatility and a measure of uncertainty or investor fear. Note the confirmation

on Fig 1 of our *a priori* expectation that *VIX* is at its peak during financial turmoil—especially during the subprime crisis—which also corresponds to periods of higher uncertainty.

Ex ante, investors demand an illiquidity risk premium for holding illiquid assets. However, ex post, one is likely to actually earn the premium when markets are liquid and to be harmed when markets are illiquid. In particular, it seems logical that liquidity is likely to dry up during recessions (for a further discussion of market liquidity in the context of regulation and the global financial crisis, see [66]).

## Estimation of the conditional version of the augmented Fama-French model

In this section, we compare the basic OLS estimation of the augmented Fama-French [1,2] model with time-varying coefficients with our proposed RIV GMM algorithm (upper panel of Tables 3 and 4). As explained in [67], a linear model with time-varying parameters like the one that we are estimating in this article is in fact a very general nonlinear model. This follows from White's theorem cited in [68].

Examining the results in Tables 3 and 4, note that the sectors (Energy and Health) with high and low sensitivity to the illiquidity factor *IML* are both significant applying OLS estimation. However, when employing our RIV GMM, the significance level for both *IML* coefficients drops but the signs of the coefficients remain the same as the corresponding OLS estimates. Except for the market factor whose coefficients are all significant at the 1% level, this comment also applies for all other factors with the GMM coefficients almost always becoming insignificant.

Comparing the two estimation approaches for the high and low beta sectors (Durables and Utilities), a different picture emerges. The significance of illiquidity increases as we move from OLS to RIV GMM. This may be indicative of specification errors in the model. Adrian et al. [3] note that standard illiquidity measures are endogenous and are simply proxies for more complex phenomena that could perhaps be modeled with an additional equation. Furthermore, when using the RIV GMM, the coefficient of *IML* is high, positive and significant for both sectors, which suggests that the stock issued by these industry sectors are quite illiquid. This argument is supported by the coefficients of first order autocorrelation for these sectors which are significant (Table 1). For both sectors, the market risk premium $\beta$ declines as we move from OLS to RIV GMM.

Interestingly, based our approach RIV GMM (upper panel of Table 4), note that the low beta sector (Utilities), which is less exposed to the market, seems to require a higher illiquidity premium on average than the high beta sector (Durables). The firms in this low beta sector generally have high debt-to-equity ratios which means that in times of crisis this leverage increases, making them more vulnerable and hence less liquid, in spite of of their low market betas. When averaging over the 12 sectors, we get a result similar to the low beta sector. That is, moving from OLS to RIV GMM, beta declines and *IML* sensitivity increases.

Turning to the parameters for the *conditional* alpha and beta models (lower panel of Tables 3 and 4), the OLS and RIV GMM estimates for the various alpha parameters $\alpha_0$, $c_1$, and $c_2$ are mainly insignificant. This is not so surprising since these portfolios are not actively managed as in the case, for example, of hedge funds where they should be actively managed to generate some positive alpha. Examining the beta model parameters—i.e., $\beta_0$, $c_3$, $c_4$, and $c_5$—tells a different story. For OLS estimation, the beta model parameter are overwhelmingly significant. The impact of the market risk premium on beta exposure is usually negative, which suggests that beta tends to increase when market conditions deteriorate. Sectors thus seem to have difficulties in controlling their beta when stock markets drop, at least in the short run. The impact

**Table 3. Parameter estimations for the conditional Fama-French model via OLS by liquidity and beta attributes.**

| Cond. model | $avg\ \alpha$ | $avg\ \beta_1$ | $\beta_2$ | $\beta_3$ | $\beta_4$ | $\beta_5$ | $\beta_6$ |
|---|---|---|---|---|---|---|---|
| | constant | $R_m$-$R_f$ | SMB | HML | RMW | CMA | IML |
| High *IML* (Enrgy) | 0.0053 | 0.9010 | -0.1328 | 0.1181 | 0.1634 | 0.3933 | 0.0874 |
| *t-stat* | *0.03* | *6.89\*\*\** | *-2.12\*\** | *1.42‡* | *2.00\*\** | *3.14\*\*\** | *1.74\** |
| Low *IML* (Hlth) | 0.2704 | 0.8543 | -0.1619 | -0.4963 | 0.3606 | 0.3454 | -0.1279 |
| *t-stat* | *0.92* | *9.76\*\*\** | *-3.70\*\*\** | *-8.58\*\*\** | *6.31\*\*\** | *3.95\*\*\** | *-3.65\*\*\** |
| High beta (Durbl) | -0.4697 | 1.2214 | 0.1762 | 0.5099 | 0.1456 | -0.0785 | 0.0608 |
| *t-stat* | *-1.28‡* | *8.46\*\*\** | *3.39\*\*\** | *7.42\*\*\** | *2.15\*\** | *-0.76* | *1.46†* |
| Low beta (Utils) | -0.0525 | 0.6278 | -0.1154 | 0.2429 | 0.1526 | 0.3505 | 0.0711 |
| *t-stat* | *-0.28* | *6.20\*\*\** | *-2.41\*\** | *3.84\*\*\** | *2.44\*\** | *3.67\*\*\** | *1.86\** |
| 12-sector avg. | -0.0635 | 0.9918 | 0.0268 | 0.0711 | 0.1851 | 0.0963 | 0.0062 |
| *t-stat* | *-1.84\** | *38.71\*\*\** | *3.14\*\*\** | *6.29\*\*\** | *16.57\*\*\** | *5.62\*\*\** | *0.90* |
| **Eqs (8) and (9)** | $\alpha_0$ | $c_1$ | $c_2$ | $\beta_0$ | $c_3$ | $c_4$ | $c_5$ |
| | constant | $R_m$-$R_f$ | Spread | constant | $R_m$-$R_f$ | Spread | IML |
| **High *IML* (Enrgy)** | | | | | | | |
| Alpha | 0.2441 | -0.0211 | -0.1368 | | | | |
| *t-stat* | *0.86* | *-0.55* | *-1.01* | | | | |
| Beta | | | | 1.0245 | -0.0192 | -0.0693 | 0.0041 |
| *t-stat* | | | | *15.39\*\*\** | *-2.64\*\*\** | *-2.38\*\** | *0.42* |
| **Low *IML* (Hlth)** | | | | | | | |
| Alpha | 0.6500 | 0.0151 | -0.2319 | | | | |
| *t-stat* | *3.28\*\*\** | *0.56* | *-2.45\*\** | | | | |
| Beta | | | | 0.7800 | -0.0150 | 0.0492 | -0.0012 |
| *t-stat* | | | | *16.79\*\*\** | *-2.97\*\*\** | *2.43\*\** | *-0.18* |
| **High beta (Durbl)** | | | | | | | |
| Alpha | -0.7500 | 0.0653 | 0.1472 | | | | |
| *t-stat* | *-3.17\*\*\** | *2.05\*\** | *1.31‡‡* | | | | |
| Beta | | | | 1.0735 | 0.0082 | 0.0911 | -0.0207 |
| *t-stat* | | | | *19.46\*\*\** | *1.37‡* | *3.78\*\*\** | *-2.56\*\** |
| **Low beta (Utils)** | | | | | | | |
| Alpha | -0.2320 | -0.0284 | 0.1159 | | | | |
| *t-stat* | *-1.07* | *-0.97* | *1.12* | | | | |
| Beta | | | | 0.7055 | -0.0139 | -0.0456 | 0.0132 |
| *t-stat* | | | | *13.90\*\*\** | *-2.52\*\** | *-2.06\*\** | *1.78\** |
| **12-sector avg.** | | | | | | | |
| Alpha | -0.0187 | 0.0016 | -0.0273 | | | | |
| *t-stat* | *-0.48* | *0.31* | *-1.48†* | | | | |
| Beta | | | | 0.9596 | -0.0017 | 0.0199 | -0.0005 |
| *t-stat* | | | | *105.85\*\*\** | *-1.72\** | *5.02\*\*\** | *-0.35* |

The t-statistics (*t-stat*) are in italics. We used two methodologies to estimate our time-varying model: OLS and RIV GMM.

Significance levels of the p-values are 1% indicated by \*\*\*; 5%, \*\*, 10%, \*; 15%, †; 20%, ‡.

of the two other factors on beta exposure—i.e., the term spread and market illiquidity—differs according to sectors. Funding liquidity, which is represented by the term spread variable parameter estimate $c_4$, is quite significant in all of our experiments. In falling financial markets, a drop in funding liquidity results in a decrease in the market beta of the Energy sector, a sector much exposed to market illiquidity. Indeed, according to Table 3, an increase of 1% in *IML*

**Table 4. Parameter estimations for the conditional Fama-French model via RIV GMM by liquidity and beta attributes.**

| Cond. model | $avg\ \alpha$ | $avg\ \beta_1$ | $\beta_2$ | $\beta_3$ | $\beta_4$ | $\beta_5$ | $\beta_6$ |
|---|---|---|---|---|---|---|---|
| | constant | $R_m$-$R_f$ | SMB | HML | RMW | CMA | IML |
| High *IML* (Enrgy) | -0.3320 | 0.8641 | -0.1782 | -0.6222 | 0.6803 | 0.1727 | 0.4344 |
| *t-stat* | -0.38 | 3.17*** | -0.52 | -0.71 | 1.17 | 0.11 | 1.60† |
| Low *IML* (Hlth) | 1.0472 | 0.4961 | 0.1669 | -0.4321 | 0.5481 | 0.4294 | -0.2729 |
| *t-stat* | 0.61 | 1.70* | 0.79 | -0.89 | 1.51† | 0.52 | -1.50† |
| High beta (Durbl) | -0.2513 | 1.1342 | -0.3397 | 1.3335 | -0.5118 | -0.3590 | 0.3996 |
| *t-stat* | -0.27 | 5.22*** | -1.06 | 2.09** | -0.94 | -0.27 | 1.78* |
| Low beta (Utils) | -0.4844 | 0.4904 | -0.0279 | -0.0540 | 0.3688 | 0.7775 | 0.8118 |
| *t-stat* | -0.48 | 4.32*** | -0.13 | -0.07 | 0.83 | 0.67 | 3.85*** |
| 12-sector avg. | 0.0660 | 0.7815 | 0.0035 | 0.1856 | 0.2008 | -0.1936 | 0.1563 |
| *t-stat* | 0.11 | 4.75*** | 0.03 | 0.85 | 1.00 | -0.37 | 1.96** |
| **Eqs (8) and (9)** | $\alpha_0$ | $c_1$ | $c_2$ | $\beta_0$ | $c_3$ | $c_4$ | $c_5$ |
| | constant | $R_m$-$R_f$ | Spread | constant | $R_m$-$R_f$ | Spread | IML |
| **High *IML* (Enrgy)** | | | | | | | |
| Alpha | 0.3030 | -0.1577 | -0.3337 | | | | |
| *t-stat* | 0.14 | -1.19 | 0.02 | | | | |
| Beta | | | | 0.5494 | -0.0309 | 0.1972 | 0.0019 |
| *t-stat* | | | | 0.65 | -0.38 | 0.74 | 0.04 |
| **Low *IML* (Hlth)** | | | | | | | |
| Alpha | 1.2108 | -0.3786 | 0.0141 | | | | |
| *t-stat* | -0.50 | -0.33 | 1.06 | | | | |
| Beta | | | | 0.4469 | -0.0638 | 0.0461 | 0.0094 |
| *t-stat* | | | | 1.07 | -1.35‡ | 0.33 | 0.30 |
| **High beta (Durbl)** | | | | | | | |
| Alpha | 1.1289 | -0.1446 | 0.5684 | | | | |
| *t-stat* | -1.38‡ | -0.24 | 0.82 | | | | |
| Beta | | | | 1.2294 | -0.0451 | -0.0432 | -0.0023 |
| *t-stat* | | | | 1.83* | -0.64 | -0.18 | -0.05 |
| **Low beta (Utils)** | | | | | | | |
| Alpha | -0.7879 | -0.2197 | 0.2468 | | | | |
| *t-stat* | -1.66* | -0.85 | 1.03 | | | | |
| Beta | | | | 0.3665 | -0.0146 | 0.0770 | 0.0064 |
| *t-stat* | | | | 0.61 | -0.17 | 0.43 | 0.12 |
| **12-sector avg.** | | | | | | | |
| Alpha | 0.0310 | -0.1345 | 0.0606 | | | | |
| *t-stat* | 0.11 | -1.02 | 0.69 | | | | |
| Beta | | | | 0.6944 | -0.0331 | 0.0595 | 0.0102 |
| *t-stat* | | | | 2.28** | -0.98 | 0.84 | 0.64 |

The t-statistics (*t-stat*) are in italics. We used two methodologies to estimate our time-varying model: labeled OLS and RIV GMM.

Significance levels of the p-values are 1% indicated by ***; 5%, **; 10%, *; 15%, †; 20%, ‡.

result in a rise of 0.08% in the Energy sector's return. Therefore, given that the Energy's return embeds an illiquidity premium, it seems relevant to assume that its funding constraint binds when market liquidity worsens. Since market beta comoves positively with leverage, the resulting deleveraging of the Energy sector following an increase in the term spread leads to a decrease in its market beta (Table 3). The Health sector, which is exposed negatively to *IML*,

adopts an opposite behavior regarding financial funding liquidity. For this sector, an increase in *IML* leads to a decrease in its stock return. Note that this reaction may be due to an expectation effect, an increase in illiquidity resulting in a further expected deterioration of this market dimension, giving rise to a drop in returns (see [54] and references therein). Since the return of the Health sector does not incorporate an illiquidity premium, it does not deleverage after a rise in the term spread. Its funding constraint is probably not binding. A decrease in funding liquidity thus results in an increase in its beta.

The high and low beta sectors—i.e., Durables and Utilities, respectively—have both funding and market liquidity ($c_4$ and $c_5$) significance. The securities issued by Utilities are more akin to bonds than to stocks while the reverse holds true for Durables. This argument is in line with the relatively low beta for the former and the relatively high beta for the latter. We may thus conjecture that Utilities is more concerned with credit risk—which is associated with our funding liquidity indicator (term spread)—than with market risk, which is associated with our market liquidity indicator (*IML*). Our findings bear out this hypothesis. When market liquidity deteriorates, the Durables sector reduces its beta—i.e., its market risk. In contrast, when funding liquidity worsens this sector does not deleverage, in the sense that it tends to let its beta increase. Its funding constraint thus does not seem to bind when market liquidity worsens. In this respect, *IML* is not significant in the return equation of this sector. Utilities follows the opposite behavior. When credit risk increases, it seems to deleverage since its market beta decreases following this kind of shock, which suggests that its funding constraint is then binding. Consistent with that, the return of Utilities embeds a significant illiquidity premium which is significant at 10% level. However, since this sector is less concerned about market risk, it does not seem to counteract a rise in its beta following an increase in *IML*. However, for the 12-sector average, only the funding liquidity is significant.

Summing up, funding liquidity and market liquidity are mainly at play in recessions or crises—i.e., *IML* and the term spread tend to increase concomitantly during crises and to decrease in tandem outside crises. Therefore, funding liquidity and market liquidity tend to deteriorate simultaneously. Sectors whose returns embed an illiquidity premium—i.e., that become illiquid during crises—are then more exposed to a binding funding constraint. They thus must deleverage in times of crises, which reduces their systematic risk as measured by the market beta. However, a sector which deleverages seems to have difficulties in controlling the impact of *IML* on its market beta. In contrast, a sector which does not deleverage after a deterioration in funding liquidity tends to take measures to reduce the impact of a rise in market illiquidity on its beta.

Turning to our RIV GMM approach, virtually all of the beta model parameters are insignificant. The conditional model may be seen as an alternative to Kalman filtering to obtain time-varying parameters for the performance measure alpha and the systematic risk measure beta. The evidence of Ferson and Schadt [3] therefore seems to be quite compelling at first glance. However, some literature (e.g., [13]) suggests that this modeling approach may lead to biased estimation and a simple static approach may be preferable. Our RIV GMM results seem therefore to be in line with this literature, but this analysis is not necessarily conclusive. Below, we continue our analysis of the time-varying performance measure alpha and the systematic risk measure beta.

The time-varying performance measure alpha computed on the average of our 12-sector portfolio is quite sensitive to the estimation approach (Fig 2A). Our RIV GMM time-varying conditional alpha seems to be much more sensitive to the business cycle. However, a clearer picture emerges when looking at the time-varying conditional beta (Fig 2B). Note that the subprime crisis is well highlighted as the beta increases more with RIV GMM than with OLS at the start of the subprime crisis but also decreases more precipitously with the IV method.

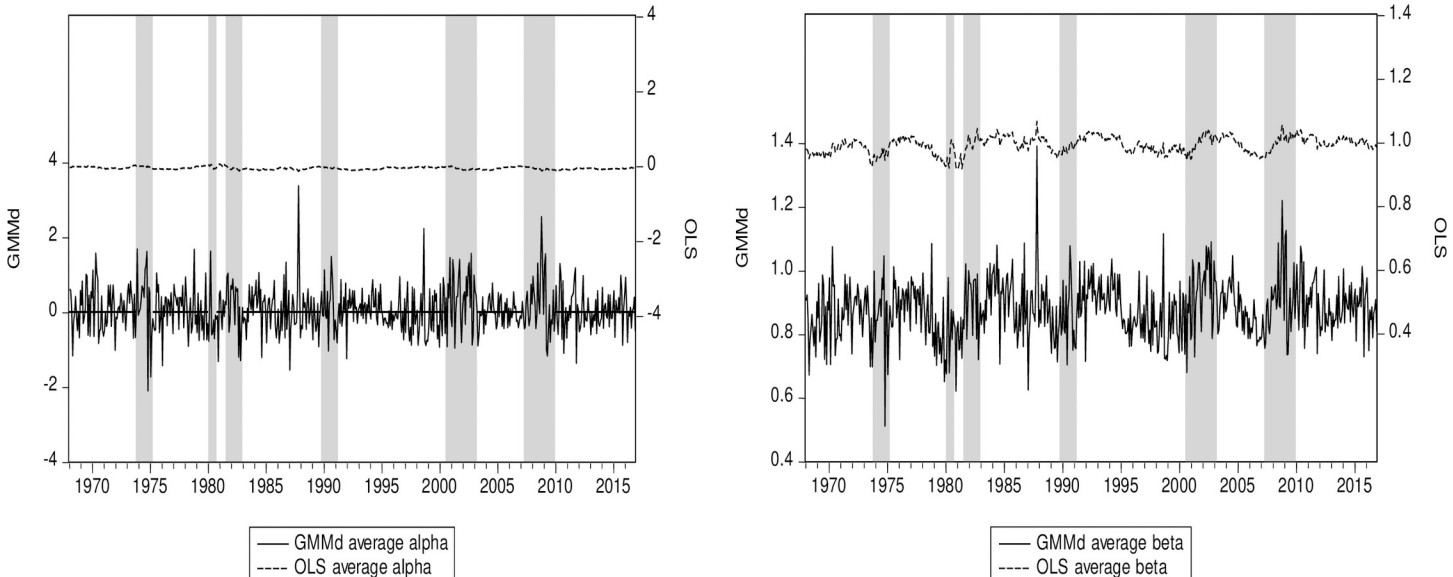

**Fig 2.** (a) 12-sector average alpha RIV GMM & OLS. (b) 12-sector average beta RIV GMM & OLS. Notes: To capture the business cycle, we use *IML* as the main driver in the dynamic Eq (9) for our cyclical beta. RIV GMM (GMMd) is our robust instrumental variables approach. Shaded areas represent US recessions or crises. The OLS and GMM series are plotted on the same scale.

Firms thus deleverage quickly during a crisis, which leads to a reduction in the beta. Note that the crisis is an extreme event that is well modelled by our designed robust IV GMM which is based on higher cross-sample moments of third and fourth degrees. It has been demonstrated [69,70] that the departure from normality can be properly reflected in these higher moments. However, this tracking of the business cycle is an unintended consequence of our design to account for endogeneity issues and/or measurement errors. In our view, this should make these instruments even more appealing to the empirical practitioner.

Continuing this line of thought, we see that the subprime crisis is much more pronounced in the energy sector than the 12-sector average (Fig 3B). The same behavior is also found in the utilities sector (Fig 4B). The reason might well be that financial practitioners working in the energy markets are likely sophisticated because they use derivatives and commodities to manage their exposures. Note also that our RIV GMM beta estimators seem to be moving in a direction opposite the OLS ones during the subprime crisis (Figs 3B and 4B). For instance—especially for the Energy sector—the beta estimated with OLS tends to decrease at the beginning of the subprime crisis, while the beta estimated with RIV GMM tends to increase. This latter result we obtain with GMM is appealing since systematic risk tends to increase at the start of a crisis. Indeed, in addition to a surprise effect, the deleveraging process, which results in decrease in the beta, takes time. The opposite behavior of the betas estimated with OLS and RIV GMM may be due to reverse causation, an issue neglected by OLS but tacked by our robust instruments. Both of the RIV GMM alphas hint at some cyclicality (Figs 3A and 4A), and while it is harder to say definitively why, the connection should not be surprising as alpha should be close to zero in efficient markets.

Turning to the health sector (Fig 5B), note that the OLS dynamic time-varying market beta seems to be higher than the RIV GMM one most of the time, implying that the the health sector firms are less exposed to the market than it initially appears. In addition to being heavily subsidized, the health industry seems to have a captive market and is less prone to cyclicality. It could even be countercyclical, since in a downturn more people may become sick due to

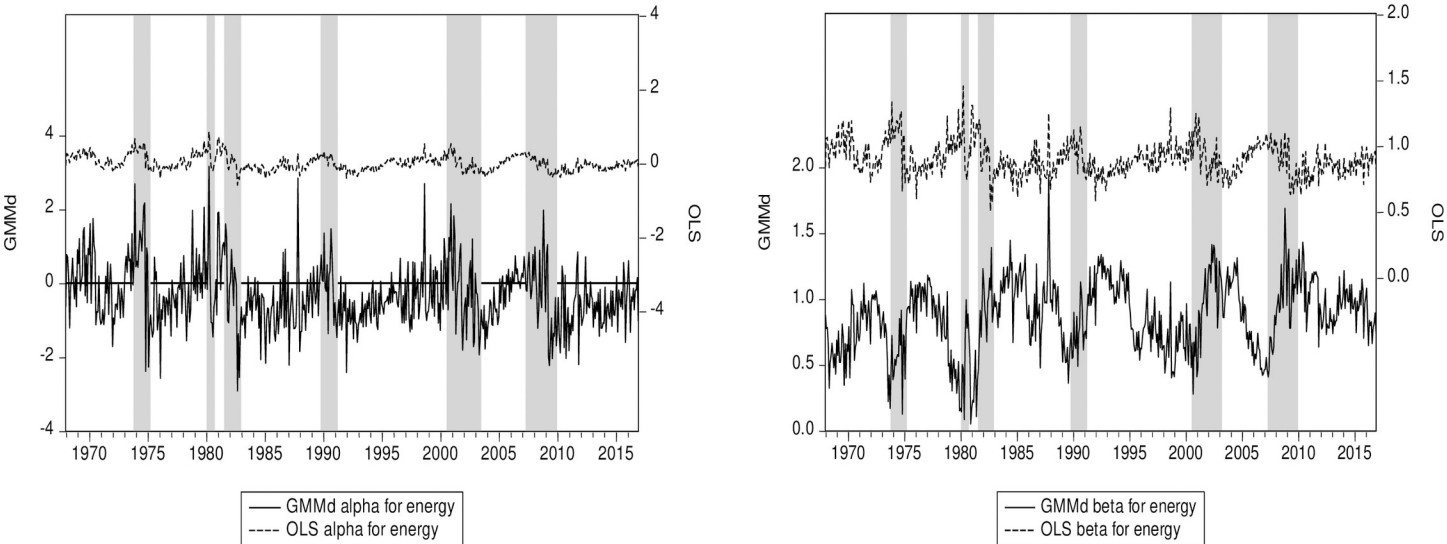

**Fig 3.** (a) Energy alpha for RIV GMM and OLS. (b) Energy beta for RIV GMM and OLS. Notes: The energy sector has the highest value of the OLS factor loading for illiquidity *IML*. RIV GMM (GMMd) is our robust instrumental variables approach. Shaded areas represent US recessions or crises. The OLS and GMM series are plotted on the same scale.

stress from job losses and other negative economic conditions and thus may call on their health services more often. In fact, the RIV GMM alpha (Fig 5A) seems to be above the OLS one during the subprime crisis. Furthermore, the RIV GMM alpha is higher on average than the OLS alpha, which is in line with the RIV GMM lower beta.

Finally, examining the durables sector (Fig 6A), we note that the RIV GMM time-varying alpha seems to be much more sensitive to the business cycle than the OLS time-varying alpha, especially during the subprime crisis where the alpha initially went negative before recovering quickly to a positive value. This suggests that the risk management techniques put in place by

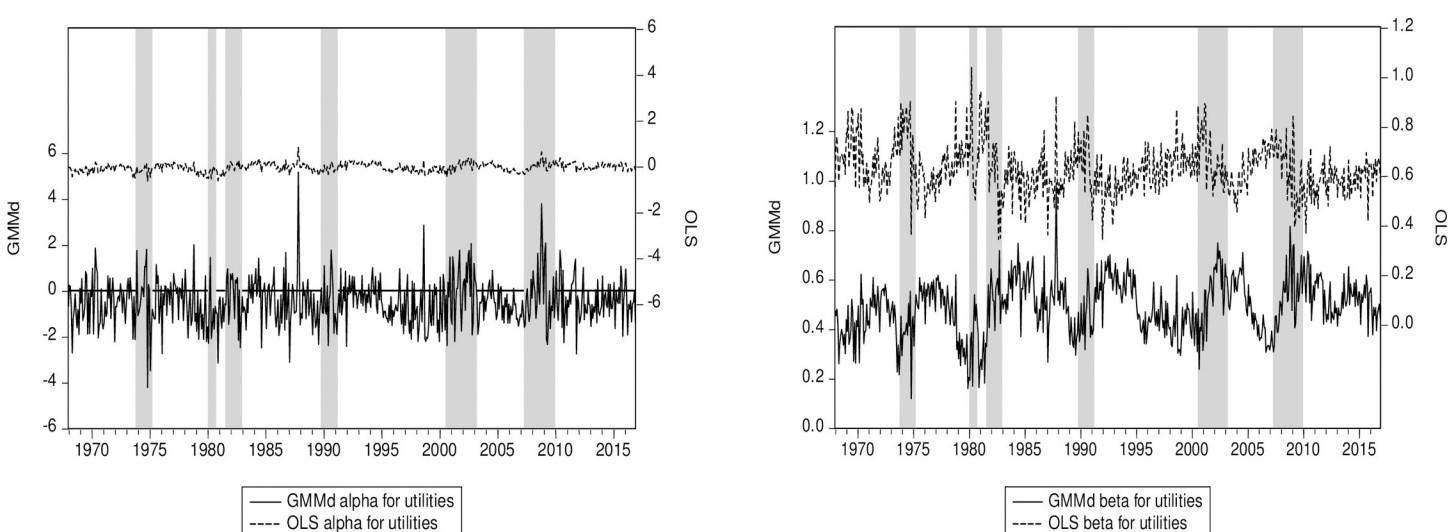

**Fig 4.** (a) Utilities alpha for RIV GMM and OLS. (b) Utilities beta for RIV GMM and OLS. Notes: This sector has the lowest OLS beta among the 12 sectors. RIV GMM (GMMd) is our robust instrumental variables approach. Shaded areas represent US recessions or crises. The OLS and GMM series are plotted on the same scale.

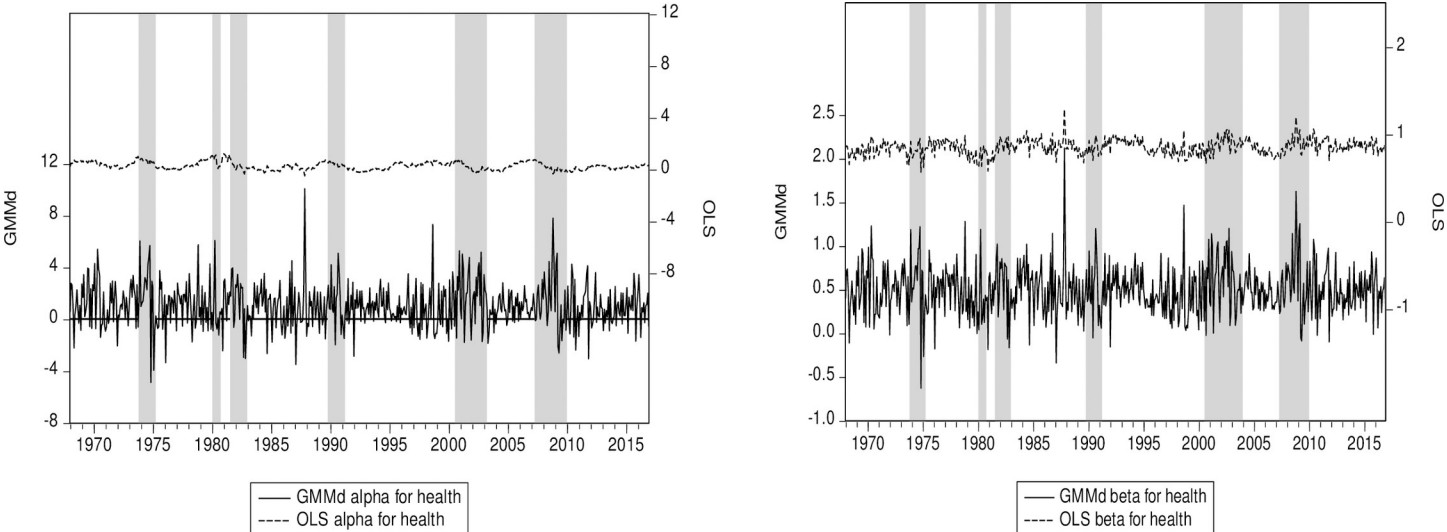

**Fig 5.** (a) Health alpha for RIV GMM and OLS. (b) Health beta for RIV GMM and OLS. Notes: The health sector has the lowest value of the OLS factor loading for illiquidity *IML*. RIV GMM (GMMd) is our robust instrumental variables approach. Shaded areas represent US recessions or crises. The OLS and GMM series are plotted on the same scale.

firms in this sector to fight the crisis—such as deleveraging—enjoy some success. Note that the same result is obtained for the aggregate of our 12 sectors, albeit less pronounced (Fig 2B). The durables sector has the highest average OLS beta. Comparing time-varying betas (Fig 6B), the RIV GMM beta seems more volatile than the OLS beta. Reverse causation is also present here, as the beta estimated by OLS and the one based on RIV GMM tend to move in opposite direction.

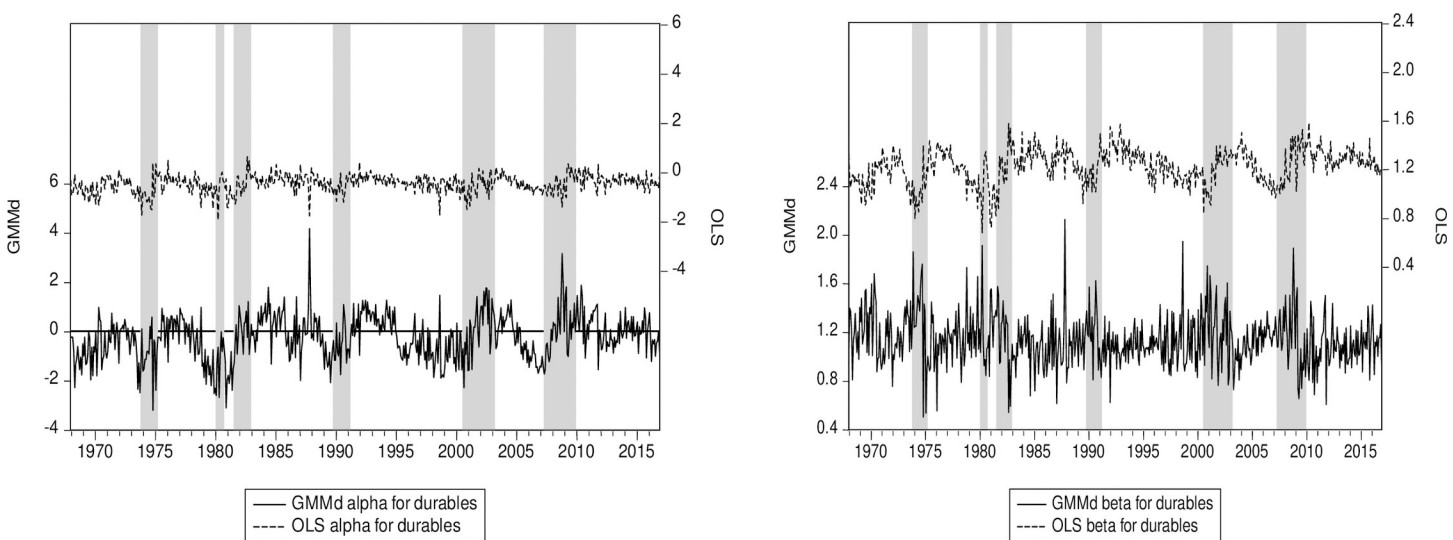

**Fig 6.** (a) Durables alpha for RIV GMM and OLS. (b) Durables beta for RIV GMM and OLS. Notes: This sector has the highest OLS beta among the 12 sectors. RIV GMM (GMMd) is our robust instrumental variables approach. Shaded areas represent US recessions or crises. The OLS and GMM series are plotted on the same scale.

## Robustness check

As a robustness check, we consider the illiquidity measure developed by Amihud [71]. This measure is the daily ratio of absolute stock return scaled by dollar volume, averaged over each month, i.e.,

$$LIQ\_Amihud = \frac{1}{D_i} \sum_{i=1}^{D_i} \frac{|R_i|}{Vol_i} \tag{21}$$

where $D_i$ is the number of days of the month, $R_i$ is the daily return on stock $i$ and $Vol_i$ is its corresponding trading volume. We compute the Amihud (2002) illiquidity measure with the S&P500. The Amihud ratio quantifies the price/return response to a given size of trade. When the Amihud ratio is high, liquidity is low.

Liquidity risk is multidimensional and more than one liquidity measure may be needed to capture different aspects of liquidity risk. For instance, Goyenko et al. [61] show that the Pástor and Stambaugh's [18] liquidity measure fails to capture the price impact of trade, while Amihud's [71,72] measure can be considered as a good proxy of this aspect of liquidity risk. To account for this multidimensionality, we first regress equation (10) by adding the Amihud ratio in this equation as an additional illiquidity factor aside *IML*. This regression allows gauging the significance of this ratio as an explanatory variable of the returns of our 12-sector portfolios and to compare it to the *IML* measure. Note that Blau and Whitby [73] have also augmented the Fama and French model with the Amihud ratio. Second, we reestimate equation (7) by substituting the Amihud ratio to *IML* in the equation of the market beta. Note that we only rely on OLS in this section because the results we obtain with GMM when using the Pástor and Stambaugh's [18] illiquidity measure are not enough conclusive.

Table 5 provides the liquidity betas of our 12 portfolios computed with OLS using the *IML* and Amihud illiquidity measures. The coefficients of the Amihud ratio are significant at the 10% level or less for six sectors (among twelve). Consistent with [61], we note that the two illiquidity measures do not rank portfolio sectors in the same way. For instance, the health sector is the lowest *IML* (-0.13) but the highest *Amihud* (1.94) one. In terms of exposures, the durables sector is second when using *IML* (0.08) and eleventh (-2.36) when using *Amihud*. According to the Amihud ratio, the health, utilities, non-durables, and money sectors have the highest positive loadings to illiquidity, while the telecom, manufacturing, durables and business equipment sectors have the lowest (negative) loadings.

Table 6 provides the estimation of the conditional alpha and market beta equations using OLS. In this estimation, we substitute the Amihud ratio to *IML* as a conditioning factor in the conditional market beta equation. Analogously to our previous estimations with *IML*, the estimated coefficients for the conditional alpha equation are mostly insignificant. According to the constant, the health and telecom. sectors stand as high performers during our sample period while the durables and money sectors are low performers. These results seem to be greatly attributable to the subprime crisis which was very detrimental to Money and Durables —two very cyclical industries. The health and telecom sectors were more resilient.

In this respect, Pástor and Stambaugh [19] contend that in periods without crises, liquidity factors are estimated with less precision. Moreover, according to these researchers, liquidity betas are relatively hard to estimate—especially during economic expansions. Indeed, liquidity shocks are close to zero in normal states, which results in noisy estimates of liquidity betas. The results obtained over our whole sample period should thus be very dependent on the subprime crisis, which is included in our sample.

Turning to the equation of the conditional market beta, we note that seven sectors out of twelve display a significant Amihud ratio. The sectors with a high *Amihud* loading in Table 5

**Table 5. OLS exposures of the 12-sector portfolios to Amihud illiquidity ratio and *IML* in the augmented Fama and French five-factor model.**

|  | *Amihud* | *IML* |
|---|---|---|
| **Nodur** | 1.42 | 0.01 |
|  | 2.33** | 0.25 |
| **Durbl** | -2.36 | 0.08 |
|  | -2.47** | 1.83* |
| **Manuf** | -0.63 | 0.06 |
|  | -1.30‡ | 2.57** |
| **Enrgy** | 0.28 | 0.09 |
|  | 0.26 | 1.74* |
| **Chems** | 0.49 | 0.02 |
|  | 0.85 | 0.74 |
| **Buseq** | -2.43 | -0.01 |
|  | -2.40** | -0.01 |
| **Telcm** | -0.42 | 0.03 |
|  | -0.50 | 0.75 |
| **Utils** | 1.58 | 0.07 |
|  | 1.60* | 1.57† |
| **Shops** | 0.77 | 0.02 |
|  | 1.19 | 0.83 |
| **Hlth** | 1.94 | -0.13 |
|  | 2.29** | -3.23*** |
| **Money** | 1.16 | -0.07 |
|  | 1.78* | -2.41** |
| **Other** | -0.18 | -0.08 |
|  | -0.35 | -3.48** |

The t-statistics (*t-stat*) are in italics. Significance levels of the p-values are 1% indicated by ***; 5%, **; 10%, *; 15%, †; 20%, ‡.

We do not report the other coefficients of the augmented Fama and French five-factor model since they are not concerned by our robustness check.

tend to decrease their market beta when the Amihud ratio increases—i.e., when illiquidity rises. For instance, the money sector, which has a liquidity beta of 1.16, decreases its market beta when illiquidity increases. The non-durable goods sector, with a liquidity beta of 1.42, displays the same behavior. However, the health sector stands as an exception. Even if its liquidity beta is estimated at 1.94, its market beta decreases when market illiquidity increases. As argued previously, this sector is very specific. Perhaps is it perceived less risky in times of rising illiquidity?

In line with [19], this relationship between the liquidity beta of a sector and the reaction of its market beta to illiquidity is very dependent on crises—especially the subprime crisis. During crises, sectors with high liquidity betas should reduce their market beta—for instance, by deleveraging—since risk is particularly high for them. We also note that this relationship tends to be tighter when the market beta—i.e., systematic risk—of the sector is higher. For instance, the money sector displays high liquidity and market betas. This sector thus bears a high level of risk during crises. It is all the more important for this sector to smooth its risk during bad times. The non-durables sector is in the same situation. The relationship between the market beta and the Amihud ratio thus tends to be driven by the level of the market beta.

**Table 6. Parameter estimations for the conditional alpha and market beta equations via OLS by using the Amihud ratio.**

| | alpha | | | market beta | | | |
|---|---|---|---|---|---|---|---|
| | *constant* | *$R_m$-$R_f$* | *Spread* | *constant* | *$R_m$-$R_f$* | *Spread* | *Amihud* |
| **Nodur** | 0.0187 | 0.0080 | -0.0555 | 1.0539 | -0.0026 | -0.0010 | -0.2804 |
| | *0.08* | *0.41* | *-0.80* | *21.81\*\*\** | *-0.72* | *-0.07* | *-3.98\*\*\** |
| **Durbl** | -0.9745 | 0.0707 | 0.1469 | 1.0562 | 0.0063 | 0.0965 | -0.0051 |
| | *-2.65\*\*\** | *2.20\*\** | *1.28‡* | *13.22\*\*\** | *1.04* | *3.96\*\*\** | *-0.04* |
| **Manuf** | -0.1028 | 0.0007 | -0.0292 | 1.0256 | 0.0042 | 0.0384 | 0.1164 |
| | *-0.53* | *0.04* | *-0.49* | *24.44\*\*\** | *1.31†* | *3.00\*\*\** | *1.90\** |
| **Enrgy** | 0.5080 | -0.0186 | -0.1508 | 0.9638 | -0.0188 | -0.0715 | 0.1246 |
| | *1.15* | *-0.48* | *-1.10* | *10.05\*\*\** | *-2.59\*\*\** | *-2.44\*\** | *0.89* |
| **Chems** | -0.0650 | -0.0368 | -0.0523 | 1.0273 | 0.0009 | 0.0397 | -0.1526 |
| | *-0.28* | *-1.83\** | *-0.73* | *20.52\*\*\** | *0.25* | *2.60\*\*\** | *-2.09\*\** |
| **Buseq** | 0.2543 | 0.0015 | 0.1036 | 0.8465 | 0.0073 | 0.0687 | 0.1633 |
| | *0.88* | *0.06* | *1.16* | *13.49\*\*\** | *1.53†* | *3.59\*\*\** | *1.78\** |
| **Telcom** | 0.5751 | -0.0117 | 0.0101 | 0.7401 | 0.0072 | 0.0485 | 0.0602 |
| | *1.79\** | *-0.42* | *0.10* | *10.64\*\*\** | *1.36‡* | *2.28\*\** | *0.59* |
| **Utils** | 0.1744 | -0.0250 | 0.0854 | 0.7598 | -0.0128 | -0.0461 | -0.0897 |
| | *0.52* | *-0.85* | *0.82* | *10.3788\** | *-2.31\*\** | *-2.06\*\** | *-0.84* |
| **Shops** | 0.0191 | -0.0043 | -0.1150 | 1.1191 | 0.0066 | 0.0374 | -0.2785 |
| | *0.07* | *-0.19* | *-1.44†* | *19.97\*\*\** | *1.55†* | *2.19\*\** | *-3.41\*\*\** |
| **Hlth** | 0.5596 | 0.0103 | -0.2236 | 0.8710 | -0.0148 | 0.0503 | -0.1735 |
| | *1.80\** | *0.38* | *-2.32\*\** | *12.92\*\*\** | *-2.89\*\*\** | *2.45\*\** | *-1.76\** |
| **Money** | -0.5491 | 0.0224 | -0.0146 | 1.2474 | -0.0033 | -0.0137 | -0.1254 |
| | *-2.29\*\** | *1.08* | *-0.20* | *23.99\*\*\** | *-0.85* | *-0.87* | *-1.65\** |
| **Other** | -0.1060 | 0.0132 | -0.0969 | 1.0865 | -0.0009 | 0.0003 | 0.0699 |
| | *-0.58* | *0.83* | *-1.71\** | *27.49\*\*\** | *-0.30* | *0.03* | *1.21* |

The t-statistics (*t-stat*) are in italics. Significance levels of the p-values are 1% indicated by \*\*\*; 5%, \*\*; 10%, \*; 15%, †; 20%, ‡.

We do not report the other coefficients of the augmented conditional Fama and French five-factor model as in Table 2A and 2B since they are not concerned by our robustness check.

## Conclusion

The idea behind this paper is to provide a time-varying framework to test illiquidity in the context of the five-factor Fama-French [1,2] model relying on a novel robust instrumental variables (RIV) algorithm to tackle endogenous illiquidity proxies (see [30] for a discussion of the endogeneity issues related to illiquidity proxies). We feature the FF model in a time-varying context using the conditional modeling approach proposed by Ferson and Schadt [3] and others [4,5,6,7,8], which we estimate with OLS and compare it to our RIV in a GMM framework.

In this time-varying *conditional* model, we generally find that the most significant factor is the market one, but illiquidity may be at play, depending on the state of the economy. We find that our suggested time-varying RIV approach is a parsimonious methodology that is well suited to highlight the cyclical variations of performance and systematic risk measures that are the focus of this article. In particular, we propose a way to divide the analysis according to the states of the economy. More precisely, we look at states of high and low illiquidity and states of high and low beta. This enables us to note that the high beta energy and the low illiquidity utilities sectors have similar time-varying beta profiles, with the OLS betas often moving in the opposite direction of the corresponding RIV GMM betas. This opposite behavior is

symptomatic of a reverse causation issue, which is overlooked by OLS but addressed with our RIV GMM. The RIV GMM alphas for both sectors are also more volatile than their OLS counterparts.

We note also that our suggested estimating approach is more suited to highlighting financial crises/economic downturns, particularly the subprime crisis, compared to the OLS approach, the results obtained with RIV GMM being much more cyclical and easier to interpret. This positive consequence may be attributable to the sheer design of our RIV GMM, which is based on a weighted average of higher moments of the explanatory variables that may be able to capture the asymmetric behavior of the financial time series (e.g., the Black [64] leverage effect) that we study in this paper.

Further research could consider international markets along the lines of FF [74] or estimating/testing for measurement errors/endogenous factors or proxies and thus improving the choice of factors in FF [75]. Another direction for future research on international markets is testing for market inefficiencies causing carry trade profitability in foreign exchange (the carry trade strategy is discussed in [76], chap.7). The time-varying coefficient methodology that we investigate should also play a part in this research along the lines of [77]. Another possible avenue would be to consider nonlinear models either with a business cycle transition variable or more generally, a smooth transition approach such as the STAR or Markov regime-switching models also accounting for measurement/specification errors or endogeneity biases. This could enhance the way to capture the business cycle and to examine the stability of the FF and liquidity parameters.

## Appendix

### Implementing the RIV GMM algorithm

In this appendix, we discuss the implementation of the time-varying nature of the *conditional* model. First, our algorithm requires stacking the data and because of the lagged values in the *conditional* model, this requires to insert some NA values at the right places. Second, the robust instrumental variables (RIV) need to be generated. To do this, we use the EViews matrix language to compute our weighted average obtained via GLS of the squared and third power deviations of the explanatory variables, which is given in equation (27) below. Third, once these robust instruments are computed (referred to below as the "*riv*" instruments), simply substitute them in the GMM formula (31) which is then optimized with respect to the vector of parameters of interest.

Our RIV can be generated as follows. Assume a standard linear regression of the form

$$Y = X\beta + \varepsilon \tag{22}$$

where $X$ is assumed to be an unobserved matrix of explanatory variables. We also assume that the matrix of observed variables is measured with normally distributed error, $X^* = X + v$. This assumption allows a parsimonious proof of the consistency of the estimators that we use to obtain the robust instruments RIV. Note, however, that the Durbin [38] and Pal [39] estimators that we use to obtain these instruments are analogous in some ways to co-skewness and co-kurtosis, thereby accounting for nonlinearities usually found in financial time series. Also, $\hat{\beta}$ is computed as

$$\hat{\beta} = \hat{\beta}_{TSLS} = (X'P_z X)^{-1} X'P_z Y \tag{23}$$

where $P_z$ is the standard "projection matrix or predicted value maker" for computing

$$P_z X = Z(Z'Z)^{-1}Z'X = Z\hat{\theta} = \hat{X} \tag{24}$$

and where $Z$ denotes the optimal combination of the Durbin [38] and Pal [39] estimators using GLS. The matrix version of these Durbin and Pal estimators may be written following [78,41],

$$\beta_D = (z_1' x)^{-1} (z_1' y) \qquad \text{(Durbin)} \qquad (25)$$

$$\beta_P = (z_2' x)^{-1} (z_2' y) \qquad \text{(Pal)} \qquad (26)$$

where $z_1 = [x_{tk}^2]$, $z_2 = z_3 - 3 Diag(x'x/T)x\iota$, $z_3 = [x_{tk}^3]$, $Diag(x'x/T) = x'x/T \bullet I_k$ are vectors with $k$ representing the number of explanatory variables, and $t$ the period subscript ($t = 1, \ldots, T$). The notation $\bullet$ is the Hadamard product. The second and third power (moments) of the *de-meaned* variables ($x$) are then computed. This is analogous to computing the second and third moments of the explanatory variables. In short, the instruments are obtained by taking the matrix of explanatory variables ($X$) in deviation from their mean ($x$). Next, we obtain the weighted estimator ($\beta_H$) by an application of the optimal generalized least squares (GLS) to the following combination [41],

$$\beta_H = \Lambda \begin{pmatrix} \beta_D \\ \beta_P \end{pmatrix} \qquad (27)$$

where $\Lambda = (C'S^{-1}C)^{-1}C'S^{-1}$ is the GLS weighting matrix, $S$ is the covariance matrix of $\begin{pmatrix} \beta_D \\ \beta_P \end{pmatrix}$ under the null hypothesis (i.e., no measurement errors), and $C = \begin{pmatrix} I_k \\ I_k \end{pmatrix}$ is a matrix of two stacked identity matrices of dimension $k$. The methodology is based on the Bayesian approach of Theil and Goldberger [75], which may lead to a Bayesian shrinking process analogous to the well-know Bayesian VAR (see e.g., [14]). This yields estimators that are more asymptotically efficient or at least as asymptotically efficient as using either only the Durbin or Pal estimators. This approach for obtaining Z is implemented in (28) below in deviation form. From (24) we extract the matrix of residuals

$$riv = X - \hat{X} = X - P_Z X = (I - P_Z) X \qquad (28)$$

In (28) the matrix of robust instruments ($riv$) can then be defined individually as

$$riv_{it} = x_{it} - \hat{x}_{it} \qquad (29)$$

The variable $riv_{it}$ may be viewed as a filtered version of the endogenous variables. It potentially removes residual non-linearities that might be hidden in $x_{it}$. The $\hat{x}_{it}$ in (29) may be obtained via a parsimonious *artificial regression approach* (e.g., see [79]) as an alternative way of obtaining our $riv$ instead of (24). This approach is designed by applying OLS on the $z$ instruments

$$\hat{x}_{it} = \hat{\gamma}_0 + z\hat{\phi} \qquad (30)$$

The $z$ are computed as explained previously. We repeat here the algorithm to compute these robust IV [80] in a more compact fashion: $z = \{z_0, z_1, z_2\}$, where $z_0 = i_T$, $z_1 = x_{\bullet}x$, and $z_2 = x_{\bullet}x_{\bullet}x$ $-3x [Diag(x'x/T)]$.

Implementing the robust instrumental variables [54] *riv*, our GMM formulation that we refer to as RIV GMM may be written

$$\arg\min_{\hat{\beta}}\{T^{-1}[riv'(Y-X\hat{\beta})]'T^{-1}W[riv'(Y-X\hat{\beta})]\} \tag{31}$$

where *W* is a weighting matrix that can be estimated using the Newey-West [36] HAC estimator.

## Acknowledgments

We thank the seminar participants of the October 2017 IAES conference held in Montreal and Bertrand Candelon for providing useful comments on a previous version of this article.

## Author Contributions

**Formal analysis:** François-Éric Racicot, William F. Rentz, David Tessier, Raymond Théoret.

**Investigation:** François-Éric Racicot, William F. Rentz, David Tessier, Raymond Théoret.

**Methodology:** François-Éric Racicot, William F. Rentz, David Tessier, Raymond Théoret.

**Supervision:** François-Éric Racicot.

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
