## [Decision Letter · Decision Letter 0]

17 Jul 2019

PONE-D-19-16971

The conditional Fama-French model and endogenous illiquidity: A robust instrumental variables test

PLOS ONE

Dear Dr. Racicot,

Thank you for submitting your manuscript to PLOS ONE. After careful consideration, we feel that it has merit but does not fully meet PLOS ONE’s publication criteria as it currently stands. Therefore, we invite you to submit a revised version of the manuscript that addresses the points raised during the review process.

Please, pay attention to the major concerns of reviewers 1 and 3 about the use of an illiquidity measure for more robustness of your analysis. I think you will find the rest of reviewer’s comments very helpfull. To conclude, we would be very grateful if you can cite any other relevant papers recently published in Plos One.

We would appreciate receiving your revised manuscript by Aug 31 2019 11:59PM. To enhance the reproducibility of your results, we recommend that if applicable you deposit your laboratory protocols in protocols.io, where a protocol can be assigned its own identifier (DOI) such that it can be cited independently in the future. For instructions see: http://journals.plos.org/plosone/s/submission-guidelines#loc-laboratory-protocols

We look forward to receiving your revised manuscript.

Kind regards,

J E. Trinidad Segovia

Academic Editor

PLOS ONE

Journal Requirements:

1. Thank you for including the following funding information within the acknowledgements section of your manuscript; "Financial support from the IPAG Business School is gratefully acknowledged. We also acknowledge financial support from the Social Sciences and Humanities Research Council (SSHRC) of Canada, grant no. 435-2019-0078."

"no"

2. Thank you for including your competing interests statement; "No"

Reviewers' comments:

Reviewer's Responses to Questions

**Comments to the Author**

1. Is the manuscript technically sound, and do the data support the conclusions?

Reviewer #1: Yes

Reviewer #2: Yes

Reviewer #3: Yes

2. Has the statistical analysis been performed appropriately and rigorously? 

Reviewer #1: Yes

Reviewer #2: Yes

Reviewer #3: Yes

3. Have the authors made all data underlying the findings in their manuscript fully available?

Reviewer #1: Yes

Reviewer #2: Yes

Reviewer #3: Yes

4. Is the manuscript presented in an intelligible fashion and written in standard English?

Reviewer #1: Yes

Reviewer #2: Yes

Reviewer #3: Yes

5. Review Comments to the Author

Reviewer #1: This article examines a conditional extended Fama-French five-factor asset pricing model using the Pástor-Stambaugh (2003) tradeable liquidity measure. This paper is interesting and well written. Interestingly, the author(s) note that in their empirical section that the Kurtosis measure is related to the Skewness one, and this is one rare article that comments on this important fact, implying that Kurtosis could be proxied via Skewness squared.

Major Comment

Since the paper only uses one measure of illiquidity, I would suggest that the author(s) consider another measure of illiquidity such as Amihud (2002) as a robustness check to their result. The author(s) could either open a new section in their paper for this check or place the check in an appendix.

Minor comments

I recommend that the author(s) cite the recent papers by Pástor-Stambaugh (2019) and Amihud (2019), where these authors restate their faith in their illiquidity measures.

The author left yellow highlighting on line 415 and 424.

References

Amihud, Y. (2019). Illiquidity and stock returns: A revisit. Working Paper, Stern School of Business, New York University.

Pástor, L., Stambaugh, R. (2019). Liquidity Risk After 20 Years. Working Paper, SSRN.

Reviewer #2: This article relies on the Ferson and Schadt’s (1996) procedure to make time-varying the coefficients alpha and market beta in the Fama and French’s (2015) five-factor model augmented with an illiquidity factor. The alpha is conditioned by the market risk premium and the term spread, whereas the market beta depends on the market risk premium, the term spread and the illiquidity factor. They experiment with OLS and their IV-GMM using robust instruments since the illiquidity factor is endogenous (Adrian et al., 2017). Their study is run on the Fama and French 12-sector portfolios using monthly returns over the 1968-2016 period. The authors analyze the portfolio responses to illiquidity using two scenarios: (i) a portfolio with high beta versus a portfolio with low beta; (ii) a portfolio which embeds an illiquidity premium versus a more liquid portfolio. They find many cases of misspecifications when using OLS rather than the IV-GMM. Moreover, in each scenario, the alphas and betas of the two portfolios have a fairly different behavior. Finally, they show that the cycles of the portfolios’ alpha and beta are more plausible when using the IV-GMM rather than OLS.

Comments

This article is well-written and interesting. I recommend its publication but I have some minor comments that should be addressed.

(i) At page 10, the authors should provide more details on the HAC matrix, which is crucial in the GMM estimation process. More precisely, they should discuss the roles played by the concepts of kernel and bandwith which are associated with the HAC matrix.

(ii) At page 6, the authors must tell why they resort to the Pástor and Stambaugh’ s (2003) illiquidity factor which they label IML (illiquid portfolio returns minus liquid portfolio returns) rather than to their two other liquidity factors—i.e., the level of aggregated liquidity (gamma), and the innovations in aggregate liquidity. There is some explanation at line 339 (p. 15) but it comes too late.

(iii) Page 6. The authors should specify why a parsimonious model is appropriate in econometrics.

(iv) Page 8. In equation (9), the authors should remove VIXt-1, which introduces confusion when reading this equation. They should only explain that there are good alternatives to IML like VIX.

(v) At page 16, the authors contend that the term spread is widely accepted in macroeconomics as a leading indicator of the future state of the economy. They should provide more explanations on the relationship between the term spread and the economic prospects.

(vi) line 175: replace “the spread between ten-year federal bonds and 90-day T bills” by “the spread between ten-year Treasury constant maturity rate and 90-day Tbill rate”. Do the same correction in the notes of Table 1b.

line 182: replace the “initial results” by the “preliminary results”.

line 288: replace to “mitigate” endogeneity by to “confront” endogeneity.

Line 342: replace “90-day Tbills” by “90-day Tbill rate”.

References

Adrian T, Fleming M, Shachar O, Vogt E. Market liquidity after the financial crisis. Annual Review of Financial Economics. 2017;9:43-83.

Fama EF, French KR. A five-factor asset pricing model. Journal of Financial Economics. 2015;116:1-22.

Ferson WE, Schadt RW. Measuring fund strategy and performance in changing economic conditions. Journal of Finance. 1996;51(2):425–461.

Pástor L, Stambaugh RF. Liquidity risk and expected stock returns. Journal of Political Economy. 2003;111:642-685.

Reviewer #3: Question 1: . Is the manuscript technically sound, and do the data support the conclusions?

The paper is technically sound as the authors use a robust instrumental variable approach to estimate a financial model based on expected values. More specifically, they apply

the GMM approach to a fixed and random effects panel data framework. They allow not only for the Jensen α performance measure to vary across sectors but also the β systematic risk measure to vary. The data and their empirical framework support the conclusion that the most significant factor is the market factor, but – interestingly - illiquidity may be at play too, depending on the state of the economy.

Question 2: Has the statistical analysis been performed appropriately and rigorously?

Yes. I would recommend, with the goal of increasing the reliability to the conclusions of the paper, to do a robustness check with Amihud's liquidity indicator (2002, 2019). In other words, this procedure would compare the paper’s results with Pastor-Stambaugh's indicator to the results that would be obtained with Amihud’s indicator.

AMIHUD, Y. (2002). “Illiquidity and stock returns: Cross-section and time-series effects”. Journal of Financial Markets, 5: 31-56.

Amihud, Yakov and Levi, Shai, “The Effect of Stock Liquidity on the Firm's Investment and Production” (April 15, 2019). Available at SSRN: https://ssrn.com/abstract=3183091.

Question 3: Have the authors made all data underlying the findings in their manuscript fully available?

The main source of data is French’s website. St. Louis Federal Reserve FRED database is source of for the ten-year Treasury bond rate and the three-month T-bill rate. The Pástor-Stambaugh tradable liquidity risk factor available is from the Pastor’s website. The implied volatility index VIX is available from the CBOE database. It will be helpful if the authors include the links to these publicly available data sources in the text.

Question 4: Is the manuscript presented in an intelligible fashion and written in standard English?

Yes. I had no problem reading and understanding the text.

6. PLOS authors have the option to publish the peer review history of their article (what does this mean?). If published, this will include your full peer review and any attached files.

Reviewer #1: No

Reviewer #2: Yes: José-María Montero

Reviewer #3: No

---

## [Author Response · Author response to Decision Letter 0]

31 Jul 2019

Response to Reviewers

Manuscript: PONE-D-19-16971

Title: The conditional Fama-French model and endogenous illiquidity: A robust instrumental variables test

Editor

Comment

Please, pay attention to the major concerns of reviewers 1 and 3 about the use of an illiquidity measure for more robustness of your analysis. I think you will find the rest of reviewer’s comments very helpfull. To conclude, we would be very grateful if you can cite any other relevant papers recently published in Plos One.

Reply

In the new robustness check section (p.27), we have re-estimated our models with the Amihud illiquidity ratio. In this respect, we cite Blau and Whitby [80, PLoS ONE] who also rely on the Amihud ratio to estimate the Fama and French model. 

Reviewer 1

Major Comment

Since the paper only uses one measure of illiquidity, I would suggest that the author(s) consider another measure of illiquidity such as Amihud (2002) as a robustness check to their result. The author(s) could either open a new section in their paper for this check or place the check in an appendix.

Reply

To address this valuable comment, we have re-estimated our models with the Amihud illiquidity ratio in our new robustness check section (p.27).

Minor comments

Comment

I recommend that the author(s) cite the recent papers by Pástor-Stambaugh (2019) and Amihud (2019), where these authors restate their faith in their illiquidity measures.

Reply

Done. See references [62] and [6].

Comment

The author left yellow highlighting on line 415 and 424.

Reply

Removed. 

Reviewer 2

Comment

(i) At page 10, the authors should provide more details on the HAC matrix, which is crucial in the GMM estimation process. More precisely, they should discuss the roles played by the concepts of kernel and bandwith which are associated with the HAC matrix.

Reply

We address this issue at pages 11-12 and provide the relevant references. 

Comment

(ii) At page 6, the authors must tell why they resort to the Pástor and Stambaugh’ s (2003) illiquidity factor which they label IML (illiquid portfolio returns minus liquid portfolio returns) rather than to their two other liquidity factors—i.e., the level of aggregated liquidity (gamma), and the innovations in aggregate liquidity. There is some explanation at line 339 (p. 15) but it comes too late.

Reply

We provide the required explanations at page 6.

Comment

(iii) Page 6. The authors should specify why a parsimonious model is appropriate in econometrics.

Reply

We answer to this comment at page 7. 

Comment

(iv) Page 8. In equation (9), the authors should remove VIXt-1, which introduces confusion when reading this equation. They should only explain that there are good alternatives to IML like VIX.

Reply

Done at pages 8 and 9. 

Comment

(v) At page 16, the authors contend that the term spread is widely accepted in macroeconomics as a leading indicator of the future state of the economy. They should provide more explanations on the relationship between the term spread and the economic prospects.

Reply

We add more details on this indicator at page 18. 

Comment

(vi) line 175: replace “the spread between ten-year federal bonds and 90-day T bills” by “the spread between ten-year Treasury constant maturity rate and 90-day Tbill rate”. Do the same correction in the notes of Table 1b.

Reply

Done. 

Comment

line 182: replace the “initial results” by the “preliminary results”.

line 288: replace to “mitigate” endogeneity by to “confront” endogeneity.

Line 342: replace “90-day Tbills” by “90-day Tbill rate”.

Reply

Done.

Reviewer 3

Comment

Question 2: Has the statistical analysis been performed appropriately and rigorously?

Yes. I would recommend, with the goal of increasing the reliability to the conclusions of the paper, to do a robustness check with Amihud's liquidity indicator (2002, 2019). In other words, this procedure would compare the paper’s results with Pastor-Stambaugh's indicator to the results that would be obtained with Amihud’s indicator.

Reply

To address this valuable comment, we have re-estimated our models with the Amihud illiquidity ratio in our new robustness check section (p.27). In this respect, see Table 3 for a comparison between the Amihud ratio and IML. We also updated the references on Amihud and Pastor-Stambaugh. 

Comment

Question 3: Have the authors made all data underlying the findings in their manuscript fully available?

The main source of data is French’s website. St. Louis Federal Reserve FRED database is source of for the ten-year Treasury bond rate and the three-month T-bill rate. The Pástor-Stambaugh tradable liquidity risk factor available is from the Pastor’s website. The implied volatility index VIX is available from the CBOE database. It will be helpful if the authors include the links to these publicly available data sources in the text.

Reply

We included the websites of our databases at pages 14 and 15.

---

## [Decision Letter · Decision Letter 1]

12 Aug 2019

The conditional Fama-French model and endogenous illiquidity: A robust instrumental variables test

PONE-D-19-16971R1

Dear Dr. Racicot,

We are pleased to inform you that your manuscript has been judged scientifically suitable for publication and will be formally accepted for publication once it complies with all outstanding technical requirements.

With kind regards,

J E. Trinidad Segovia

Academic Editor

PLOS ONE

Additional Editor Comments (optional):

Reviewers' comments:

Reviewer's Responses to Questions

**Comments to the Author**

1. If the authors have adequately addressed your comments raised in a previous round of review and you feel that this manuscript is now acceptable for publication, you may indicate that here to bypass the “Comments to the Author” section, enter your conflict of interest statement in the “Confidential to Editor” section, and submit your "Accept" recommendation.

Reviewer #1: All comments have been addressed

Reviewer #2: All comments have been addressed

Reviewer #3: All comments have been addressed

2. Is the manuscript technically sound, and do the data support the conclusions?

Reviewer #1: Yes

Reviewer #2: Yes

Reviewer #3: Yes

3. Has the statistical analysis been performed appropriately and rigorously? 

Reviewer #1: Yes

Reviewer #2: Yes

Reviewer #3: Yes

4. Have the authors made all data underlying the findings in their manuscript fully available?

Reviewer #1: Yes

Reviewer #2: Yes

Reviewer #3: Yes

5. Is the manuscript presented in an intelligible fashion and written in standard English?

Reviewer #1: Yes

Reviewer #2: Yes

Reviewer #3: Yes

6. Review Comments to the Author

Reviewer #1: I am satisfied with the answers and the updated version of the manuscript. It is good for publication since the authors have updated the manuscript according to my requests.

This is a fine piece of research that should make the readership of Plos one enjoying the article.

Reviewer #2: Accept as it is

Al my comments were included in the revised

manuscript. In my opinion the current version of the paper is of interetgor the reader of the journal

In my

Reviewer #3: (No Response)

7. PLOS authors have the option to publish the peer review history of their article (what does this mean?). If published, this will include your full peer review and any attached files.

Reviewer #1: No

Reviewer #2: Yes: José-Maria Montero

Reviewer #3: No

---

## [Editor Report · Acceptance letter]

4 Sep 2019

PONE-D-19-16971R1 

The conditional Fama-French model and endogenous illiquidity: A robust instrumental variables test 

Dear Dr. Racicot:

I am pleased to inform you that your manuscript has been deemed suitable for publication in PLOS ONE. Congratulations! Your manuscript is now with our production department. 

With kind regards,

on behalf of

Dr. J E. Trinidad Segovia 

Academic Editor

PLOS ONE